



# A Waveform Skewness Index for Measuring Time Series Nonlinearity and its Applications to the ENSO-Indian Monsoon Relationship

Justin Schulte[1]*, Frederick Policelli[2], and Benjamin Zaitchik[3]

[1] Science Systems and Applications, Inc.

[2] NASA Goddard Space Flight Center

[3] Johns Hopkins University, Department of Earth and Planetary Sciences

*correspondence to*: Justin Schulte (jschulte972@gmail.com)

**Abstract**

Many geophysical time series possess nonlinear characteristics that reflect the underlying physics

of the phenomena the time series describe. The nonlinear character of times series can change with time,
so it is important to quantify time series nonlinearity without assuming stationarity. A common way to
quantify the time-evolution of time series nonlinearity is to compute sliding skewness time series, but it is
shown here that such an approach can be misleading when time series contain periodicities. To remedy this
deficiency of skewness, a new waveform skewness index is proposed for quantifying local nonlinearities

embedded in time series. A waveform skewness spectrum is proposed for determining the frequency
components that are contributing to time series waveform skewness. The new methods are applied to the
El Niño/ Southern Oscillation (ENSO) and the Indian monsoon to test a recently proposed hypothesis that
states that changes in the ENSO-Indian Monsson relationship are related to ENSO nonlinearity. We show
that the ENSO-Indian rainfall relationship weakens during time periods of high ENSO waveform skewness.

The results from two different analyses suggest that the breakdown of the ENSO-Indian monsoon
relationship during time periods of high ENSO waveform skewness is related to the more frequent
occurrence of strong central Pacific El Niño events, supporting arguments that changes in the ENSO-Indian
rainfall relationship are not solely related to noise.

## 1. Introduction

Many geophysical time series such as the solar cycle (Rusu, 2007), Quasi biennial Oscillation
(QBO; Hamiliton and Hsieh, 2002; Lu et al., 2009), and El Niño/ Southern Oscillation (ENSO;




Timmermann, 2003) are nonlinear. From a time series analysis perspective, the nonlinearities in the time series manifest as the tendency for the time series to rise faster than they fall or as the propensity for positive deviations above a horizonal axis (zero axis in case of zero-mean time series) to be greater than negative deviations below the same horizonal axis. Understanding these nonlinear time series features is important

because the nonlinear characteristics of the time series reflect the underlying physics of the phenomena in question. In the case of ENSO, the tendency for El Niño events to be stronger than La Niña events (i.e. ENSO asymmetry) is related to the propagation characteristics of equatorial Pacific SST anomalies and nonlinear dynamical heating (NHD; An and Jin,2004; Santoso et al., 2013), where strong El Niño events are associated with eastward propagating SST anomalies and enhanced NDH. On the other hand, weak

ENSO events are associated with westward propagating SST anomalies and minimal NDH. Understanding ENSO nonlinearity is also important because it is related to ENSO diversity (Duan et al., 2017) and the associated diversity of teleconnection responses.

Another reason why quantifying time series nonlinearity is important is that changing time series nonlinear characteristics is related to fluctuating time-domain correlations between two time series. As

shown by Schulte et al. (2020), the well-documented weakening ENSO-Indian monsoon relationship (Kumar et al., 2006) around the 1970s could be related to the transition of ENSO from a linear regime to a more nonlinear regime. More specifically, they showed that the ENSO-Indian monsoon relationship weakens during time periods when ENSO evolves more nonlinearly because ENSO nonlinearity contributes to the occurrence of distinct ENSO flavors (Johnson, 2013) that differentially influence the Indian summer

monsoon (Fan et al., 2017). Thus, it may not be a coincidence that ENSO transitioned to a nonlinear regime (An, 2004; An and Jin 2004; An, 2009) around the same time the ENSO-Indian monsoon began to weaken in the 1970s (Kumar et al., 1999). The results from that study oppose those of other studies that suggest that changes in the ENSO-Indian monsoon relationship are related to noise (Gershunov et al., 2001), statistical under sampling (Cash et al., 2017), Indian Ocean Dipole (Ashok et al., 2001; Ashok et al., 2004

), and Atlantic SSTs (Kuckarksi et al., 2007; Kucharksi et al., 2009; Chen et al., 2010). However, Schulte et al. (2020) did not relate ENSO nonlinearity directly to ENSO flavors and Indian rainfall during the summer monsoon season, warranting an additional study that examines the possible relationship between ENSO nonlinearity and the ENSO-Indian monsoon relationship.

Recognizing the importance of understanding nonlinear time series characteristics, many

researchers have quantified the nonlinearity of ENSO using a variety of approaches. Commonly, traditional skewness is used to measure ENSO nonlinearity (Burgers and Stephenson, 1999) because it captures the propensity for El Niño events to be stronger than La Niña events (An and Jin, 2004; An, 2004). One drawback of this skewness metric, however, is that it measures the skewness of a distribution of ENSO





index values and does not measure the skewness of specific El Niño or La Niña events. Another metric called the maximum potential intensity index proposed by An and Jin (2004) is a proxy for event skewness but with two caveats. The first caveat is that index only quantifies the amplitude of an event and therefore cannot distinguish two events that have the same amplitude but differing nonlinear characteristics. The

second caveat is that the index can only be applied to ENSO and not to arbitrary geophysical time series. Given these deficiencies, there is a clear need to construct a quantity that can measure the skewness of individual time series events regardless of the chosen study topic.

Another approach to quantifying time series nonlinearity is Fourier or wavelet-based higher-order spectral analysis. Using these methods, the cycle geometry of time series and the frequency components

contributing to time-domain nonlinearity can be quantified. For example, Schulte et al., (2020) used the methods to show how the nonlinear character of ENSO has evolved from 1871 to 2016, with ENSO being especially nonlinear in recent decades. In an earlier study, Timmermann (2003) applied Fourier-based bispectral methods to identify quadratically phase-dependent oscillators embedded in ENSO time series. While these approaches can quantify time series nonlinearity, they cannot measure the nonlinearity of

individual time series events like the other methods mentioned above.  These limitations further highlight the need to develop a method that can quantify time series event skewness.

In this study, we develop a nonlinear index that can be used measure the nonlinearity of specific events embedded in arbitrary time series. More specifically, the three objectives of the paper are the follows: (1) create a waveform skewness index to quantify local nonlinearity of time series; (2) demonstrate the

importance of the index through the application of the waveform skewness index to ENSO time series; (3) test the hypothesis that ENSO nonlinearity is related to the weakening ENSO-Indian monsoon relationship, contributing to the current debate regarding the mechanism behind the ENSO-Indian monsoon relationship changes.

## 2. Data

The Niño 1+2, Niño 3, Niño 3.4, and Niño 4 indices (available at: https://www.esrl.noaa.gov/psd/gcos_wgsp/Timeseries/) were used to measure the strength and evolution of ENSO from 1871 to 2016. These indices were calculated using the Hadley Centre Global Sea Ice and Sea Surface Temperature (HadISST1; Rayner et al., 2006) data product. The seasonal cycles were removed from the time series by subtracting the 1871-2016 monthly means from the corresponding monthly values.

After removing the seasonal cycle, the time series were standardized by dividing them by their respective standard deviations. In addition to the full time series, two seasonal time series were also considered. The first season considered was the June-July (JJ) season, which was referred to as the early monsoon season.




The second season considered was the August-September (AS) season or the late monsoon season. The JJ Niño 3 (Niño 1+2, Niño 3.4, etc.) times series was created simply averaging the June and July Niño 3 (Niño 1+2, Niño 3.4, etc.) indices and the AS was created by averaging the August and September Niño 3 (Niño 1+2, Niño 3.4, etc.) indices.

Also considered in this study was the trans Niño index, which quantifies the SST gradient across the equatorial Pacific (Trenberth and Stephenaik, 2001). The TNI was used in this study because the trans-Nino pattern has been implicated has an SST pattern contributing to changes in the ENSO-AIR relationship (Kumar et al., 2006).We defined the trans Niño index as the standardized Niño 4 minus the standardized Niño 1+2 index, contrasting with the original definition in which a 5-month running mean was applied to

the difference between the Niño 4 and Niño 1+2 indices. Our choice to forgo the smoothing step was made because seasonal time series in this study were based on 2-month seasons and a 5-month running mean would render it difficult to extrapolate seasonal relationships.

The All-India rainfall (AIR; Parthasarathy et al. 1994) time series was used to characterize changes in the Indian summer monsoon system. The AIR time series was created by averaging representative rain

gauges at various locations across India. To remove the annual cycle, the AIR time series was converted into anomaly time series by subtracting the 1871-2016 long-term mean for each month from the individual monthly values. The AIR anomaly (AIR, hereafter) time series were subsequently standardized by dividing it by its 1871-2016 standard deviation. An early (JJ) monsoon season and late monsoon (August-September) season time series were constructed in the same way as they were created for the ENSO time series.

**3. Methods**

**3.1. Waveform Skewness**

The focus of this study were quadratic phase nonlinearities that give rise to time series skewness. Quadratic nonlinearities were associated with quadratic phase dependence among oscillators with periods $P_1$, $P_2$, and $P_3$ and phases $\phi_1$, $\phi_2$, and $\phi_3$ satisfying

$$\frac{1}{P_3} = \frac{1}{P_1} + \frac{1}{P_2} \tag{1}$$

and

$$\phi_3 = \phi_1 + \phi_2. \tag{2}$$

The type of waveform resulting from quadratic nonlinearities can be measured using the bi-phase (King, 1996; Schulte, 2016, Maccarone, 2013), which is defined as

$$\psi = \phi_3 - \phi_1 + \phi_2. \tag{3}$$





A bi-phase of zero means that the quadratic nonlinearity results in positively skewed waveforms whose positive deviations from a horizonal axis are larger than the negative ones. On the other hand, waveforms associated with -180 biphase are characterized by larger negative deviations than positive ones. The bi-phase can also have values of -90 and 90, representing situations in which a time series rises faster than it falls (Schulte, 2016). This situation is not considered in this study because ENSO skewness is the focus of the paper.

The bi-phase is closely linked to the skewness of a distribution, which was computed using

$$skewness = \frac{\frac{1}{N}\sum_{i=1}^{N}(x_i - \bar{x})^3}{S}, \tag{4}$$

where $S$ in the standard deviation of a time series $x_1, x_2, \ldots, x_N$ with mean $\bar{x}$. Positive (negative) skewness meant that the right (left) tail of the time series distribution was longer than the left (right) one, reflecting the tendency for positive (negative) time series events (i.e. anomalies) to be more intense than negative (positive) ones.

To measure the time-evolution of skewness, we first partitioned a time series into overlapping segments and then computed the skewness for each individual segment. The segment length used in the calculations had to be chosen in advance, meaning that the results of the analysis depended on the chosen segment length. To see the segment length dependence, a sliding skewness analysis was applied to

$$X_1(t) = A\cos\left(\frac{2\pi}{P_1}t\right), \tag{5}$$

where $A$ is amplitude and $P_1 = 8, 16, 32$ is period (Figure 1a).

In this situation, an appropriate measure of quadratic nonlinearity was constrained to be zero and constant because the cosine function is linear and stationary. Yet, Figure 2a shows that skewness is a function of time and $P_1$ for a fixed segment length. In some cases, the skewness fluctuates between negative and positive values, which would suggest a change in the biphase even though the time series is stationary. It also appears that the cosine function with $P_1 = 16$ tends to have greater skewness than the other cosine functions even though the cosine function is linear regardless of the period. These results imply that fluctuations in skewness could be erroneously deemed as changes in time series nonlinearity in the presence of periodicities, especially for time series with low-frequency variability. Thus, when comparing time series, it could be difficult to know if one time series is truly more nonlinear than another.

It was also found that periodicities will also impact how one can interpret the skewness of truly nonlinear time series. For example, we considered the nonlinear time series given by




$$X_2(t) = cos\left(\frac{2\pi}{P_1}t + \phi_1\right) + \gamma(t)cos\left(\frac{2\pi}{P_3}t + \phi_3\right) \qquad (6)$$

where $2P_3 = P_1$, $\phi_1 = 0$, $\phi_3 = 2\phi_1$ so that sum rules (1) and (2) are satisfied. The quantity $\gamma(t)$ was called a nonlinear coefficient because as $\gamma(t)$ approached unity, positive deviations from the mean became progressively larger than negative deviations (Schulte, 2016). In this example (Example 2), $\gamma(t) = 0.8$, meaning that the degree of nonlinearity was constant. We considered the situations when $P_1 = 32$, $P_1 = 16$, and $P_1 = 8$ to understand how skewness is impacted by periodicities of varying frequencies.

As shown in Figure 1b, the quadratically phase coupled oscillators composing $X_2(t)$ give rise to positively skewed waveforms whose positive excursions are larger than the adjacent negative ones. Although this time series is stationary and nonlinear, its skewness is a function of time and $P_1$ (Figure 2b), implying that skewness is not always a consistent measure of quadratic nonlinearity. At some time points, the skewness associated with the case $P_1 = 32$ is close to zero, giving the false impression that $X_2(t)$ is linear during some time periods.

### 3.2 Waveform skewness

The deficiencies of traditional skewness motivated the construction of a new waveform skewness index that was more weakly influenced by periodicities. The construction of the waveform skewness index was also motivated by Figure 1b that shows how positive deviations are larger than negative ones in the case of zero biphase. The waveform skewness index was constructed as follows. First standardized anomaly time series were decomposed into positive and negative events using an event decomposition approach (Schulte and Lee, 2019), where positive (negative) events are contiguous strings of positive (negative) anomalies. The peak intensity of a positive (negative) event was defined as the maximum (minimum) value obtained by the data points associated with the event. The waveform skewness of a positive time series event with peak ntensity $I^p$ was then defined as

$$s_p = I^p + \frac{(I_n^a + I_n^s)}{2}, \qquad (7)$$

where $I_n^a$ is the peak intensity of the antecedent negative event, and $I_n^s$ is the peak intensity of the subsequent negative event. The waveform skewness index for negative events was calculated by multiplying a times series by -1, computing the waveform skewness indices of all fictitious positive events, and multiplying the resulting waveform skewness indices by -1.

Using the waveform skewness index, a waveform skewness time series was created by assigning to each time point the waveform skewness of the event to which the time point belongs. Performing this step for positive and negative events resulted in a waveform skewness time series whose length was nearly



equal to that of the original time series, where the length inequality occurred because the waveform skewness index of events at the end of the time series could not be computed. By construction, the waveform skewness index measured time series asymmetry with respect to a horizonal axis at a moment in time.

Unlike traditional skewness, the sliding waveform skewness time series for the linear cosine time
series shown in Figure 1a is zero and independent of time (Figure 2a). The waveform skewness of zero is consistent with how the time series is stationary and linear so that waveform skewness index is a more appropriate measure of quadratic nonlinearity in this situation. Similarly, the sliding waveform skewness time series corresponding to the nonlinear time series shown in Figure 1b are nearly constant and always positive (Figure 2b), reflecting the constant biphase of 0 and the constant degree of nonlinearity. The index
appears to be slightly dependent on the period of the cosine function, though the dependence is not as strong as it is for traditional skewness. Thus, the waveform skewness index is a more theoretically consistent measure of quadratic nonlinearity given that these nonlinearities are less impacted by periodicities. Away from the edges of the time series, the siding waveform skewness is not impacted by the chosen segment length (not shown), contrasting with sliding skewness time series whose depiction of nonlinearity depends
on what segment length is chosen.

### 3.3 Waveform Skewness Spectrum

Although the waveform skewness index measures local nonlinearity, it cannot determine the frequency components of the time series that are contributing to the time-domain waveform skewness. This frequency information is important because the frequency components determine how often positively or
negatively skewed events will occur, as Figure 1b suggests.

To determine the frequency components that are contributing to waveform skewness, we computed the waveform skewness of nonlinear modes embedded in time series, resulting in a waveform skewness spectrum. Following Schulte et al. (2020), a nonlinear mode was defined as the sum

$$X_{nonlinear} = X_{P_1} + X_{P_2} + X_{P_3} \tag{8}$$

if all the periods are unequal and as the sum

$$X_{nonlinear} = X_{P_1} + X_{P_3} \tag{9}$$

If $P_1 = P_2$, where it was assumed that the sum rules in Eqs. (1) and (2) are satisfied. Each $X_{P_i}$ were constructed using the three-step wavelet method implemented by Schulte et al. (2020). The first step of the wavelet method involved the wavelet transformation of $X(t)$ that produced an array of wavelet coefficients
containing time-frequency information about $X(t)$. In the second step, all wavelet coefficients except those




at $P_1$, $P_2$, and $P_3$ were set to zero. Computing the waveform transformation of the resulting wave coefficients yielded $X_{nonlinear}$ (Step 3). For $X_2(t)$ shown in Figure 1b, the nonlinear mode is the sum of the two cosines.

After the computation of all possible nonlinear modes, the global waveform skewness of $X(t)$ was

computed as

$$\mathcal{S}(P_1, P_2) = \frac{\sum \mathcal{S}_p^{P_1,P_2} + \sum \mathcal{S}_n^{P_1,P_2}}{N_p + N_n}, \tag{10}$$

where $\mathcal{S}_p^{P_1,P_2}$ was a waveform skewness index of a positive $X_{nonlinear}$ event, $\mathcal{S}_n^{P_1,P_2}$ was a waveform skewness index of a negative $X_{nonlinear}$ event, and the sum was computed across all $N_p$ positive events and $N_n$ negative events associated with $X_{nonlinear}$. Repeating the calculations for all nonlinear modes

resulted in the global waveform skewness spectrum.

The global waveform skewness represented the average waveform skewness index of a nonlinear mode. A positive value meant that a nonlinear mode was positively skewed, and a negative value indicated that a nonlinear mode was negatively skewed. In other words, positive (negative) values implied that the nonlinear mode was contributing to the positive (negative) waveform skewness of $X(t)$.

It was found that even realizations of a red noise process had large global waveform skewness even though they are linear. Thus, it was necessary to implement statistical significance tests. The statistical significance of global waveform skewness was assessed using a two-sided t-test, where the null hypothesis was that the global waveform skewness is equal to zero. Because global waveform skewness is the average waveform skewness associated with individual time series events, auto-correlation did not pose a challenge

for statistical significance testing.

The global waveform skewness spectrum is like the auto-bicoherence spectrum used by Schulte et al. (2020) but with a few notable differences. Unlike auto-bicoherence, high values of global waveform skewness will only occur for skewed waveforms defined with respect to a horizontal axis. On the other hand, auto-bicoherence can be high (close to 1) even if there is no skewness because the method

detects waveforms that are asymmetric with respect to a vertical and horizontal axis.

The second difference is that statistical significance of global waveform skewness can be assessed using a two-sided t-test, whereas Monte Carlo methods are required for auto-bicoherence (Schulte, 2016), rendering statistical significance testing of auto-bicoherence slow and inefficient. Another notable difference is that time-evolution of waveform skewness (i.e. local waveform skewness) corresponding to

$(P_1, P_2)$ is easier to compute than the time-evolution of local auto-bicoherence. In local auto-bicoherence



calculations, a smoothing operator is needed to create a time series that measures the degree of quadratic phase dependence (Schulte, 2016). The smoothing operation depends on wavelet scale and there are three possible scales to chosen from for a given nonlinear mode. Thus, the degree of nonlinearity will change based on the chosen scale. This problem is avoided in local waveform skewness calculations because the

calculation of waveform skewness is done in the time domain. This problem is also avoided with traditional skewness, but traditional skewness calculations are influenced by periodicities, which is problematic because statistically significant nonlinear modes contain phase-locked periodic components that allow the biphase to be relatively stable (i.e. high auto-bicoherence).

To better understand what global wavelet waveform spectrum measures, we considered the

nonlinear and nonstationary time series given by

$$X_3(t) = \frac{X_2(t)}{\sigma_2} + \frac{W(t)}{n\sigma_w},$$  (11)

where $\sigma_2$ is the standard deviation of $X_2(t)$, $W(t)$ is a realization of a white noise process with standard deviation $\sigma_w$, and $n$ is a real number representing the signal-to-noise ratio. Unlike in Example 2, we let

$$\gamma(t) = \left(\frac{t}{100}\right)^2$$  (12)

so that the nonlinearity of $X_3(t)$ increased in time. In this case, $n = 0.8$, $\phi_1 = 0$, $P_1 = 32$, and $P_3 = 16$ (Figure 3a). The global waveform skewness spectrum for this time series was then compared to the corresponding auto-bicoherence spectrum, which was obtained using the Morlet wavelet with angular frequency equal to 6. The auto-bicoherence varied from 0 to 1, where a value of 1 indicated the strongest possible degree of phase dependence among oscillators satisfying the sum rules in Eq. (1) and Eq. (2). The

readers are referred to Schulte (2016) and Schulte et al. (2020) for more details.

As shown in Figure 4a, the global waveform skewness is statistically significant around the point (32, 32), which indicates that there is quadratic phase dependence between oscillators with periods of $P_1 = 32$ and $P_3 = 16$. These statistically significant global waveform skewness values are positive so that the quadratic phase dependence is contributing to the positive skewness of $X_3(t)$ seen in Figure 3a. The auto-

bicoherence spectrum shown in Figure 4b also confirms that oscillators are contributing to the nonlinearity of $X_3(t)$. There are other points in the auto-bicoherence spectrum that are associated with statistical significance, but those points are associated with Type 1 errors. Furthermore, the corresponding points in the waveform spectrum are not associated with statistical significance, reducing confidence that the corresponding nonlinear modes are distinguishable from background noise.





The local waveform skewness time series corresponding to $X_{nonlinear}^{32,32}$ indicates that the skewness of the nonlinear mode generally increases with time (Figure 3b), consistent with how the nonlinear coefficient increases with time. Thus, the nonlinear mode is contributing to the positive waveform skewness of $X_3(t)$ to an increasing degree.

## 4. Applications to ENSO and the Indian Monsoon

### 4.1 ENSO Indices and Their Skewness

As shown in Figure 5, the ENSO time series comprise fluctuations of various magnitudes. For both the Niño 1+2 and Niño 3 indices, the most intense warm events are the 1982/1983 (Quinn et al., 1987) and 1997/1998 (McPhaden, 1999; Shen and Kimoto, 1999; Slingo and Annamalai, 2000) El Niño events. Surrounding many of the Niño 1+2 and Niño 3 warm events are cold events of lesser magnitude, reflecting the nonlinear character of ENSO (Timmermann, 2003). Other strong warm events are seen to have occurred around 1878 and 1889, but a relatively strong cold event appears after the 1889 event for the Niño 3 index, suggesting that this event is not as skewed as the 1982/1983, 1997/1998, and 2015/2016 events.

The waveform skewness time series associated with the ENSO time series better illustrate the temporal changes in nonlinearity. As shown in Figure 5c, the two most skewed Niño 1+2 events are the 1982/1983 and 1997/1998 El Niño events. In contrast, the Niño 3 index time series comprises three strongly positive skewed events located around 1982/1983, 1997/1998, and 2015/2016 (Figure 5d). The occurrence of these skewed events is consistent with how ENSO began to evolve more nonlinearly after the 1970s (Santoso et al., 2013). Despite the high intensity of the 1889 Niño 3 warm event (Figure 5b), its waveform skewness is low, implying that there is no one-to-one relationship between event intensity and waveform skewness.

The transition of ENSO from a linear regime to a nonlinear one is evident from an inspection of the (standardized) sliding skewness and sliding waveform skewness time series. As shown in Figures 6 and 7, the skewness and waveform skewness of the Niño 3 and Niño 1+2 indices are strongly positive around the 1980s and 1990s. The 10-year sliding skewness and waveform skewness time series also depict an abrupt decline in ENSO nonlinearity after the 1990s. For both ENSO indices, this decline is seen in the 20-year sliding waveform skewness time series (Figures 7a and 7b) but not in the 20-year sliding skewness time series, suggesting more uncertainty in the extent of ENSO nonlinearity after the 1990s. However, Schulte et al. (2020) found the auto-bicoherence of the ENSO indices to decline after a peak in auto-bicoherence around 1980s and 1990s so that the increase skewness could reflect the inability of skewness to always capture quadratic nonlinearities (Section 3).





The 1940s and 1950s appear to correspond with relatively low skewness and waveform skewness. In fact, the 10-year sliding skewness and waveform skewness time series associated with the Niño 1+2 index are negative around 1940 (Figure 6a). The Niño 3 index is also associated with negative waveform skewness, but the negative waveform skewness occurs around the 1950s and 1960s, a time when strong warm Niño 3 events are absent (Figure 5b). Prior to the 1940s, there is uncertainty regarding the nonlinearity of ENSO given the differences in the sliding skewness and waveform skewness time series. For example, the 10-year sliding skewness time series suggests that the Niño 1+2 index is nonlinear in the 1880s (Figure 6a), whereas the 10-year sliding waveform skewness time series suggests that the Niño 1+2 index is linear during that time.

The waveform and auto-bicoherence spectra indicate that the nonlinearity of both ENSO indices is mainly the result of quadratic phase dependence among oscillators with periods ranging from 24 to 64 months (Figure 8). For example, a statistically significant peak is located at (60,60) in both spectra for the Niño 3 and Niño 1+2 indices, indicating that phase dependence between modes with periods of 30 and 60 months is contributing to the skewness and waveform skewness seen in Figures 6 and 7. The inferred cyclic behavior of this nonlinear mode implies a tendency for positively skewed ENSO events to occur every 60 months. For the Niño 1+2 index, there is also statistically significant peak at (62,44), which means that phase-dependent oscillators with periods of 26, 44, and 62 months are contributing to Niño 1+2 index skewness seen in Figures 6a and 7a.

## 4.2 ENSO Relationship with AIR Anomalies

As shown in Figure 9, the relationship between seasonally averaged Niño 3 time series and AIR anomalies fluctuations on interdecadal timescales. From 1871 to 1970, the Niño 3-AIR relationship for the JJ and AS seasons is negative, consistent with the well-established idea that El Niño events are associated with Indian monsoon failures. However, after 1970, the AS Niño 3-AIR relationship weakens and becomes nearly positive. The weakening is not seen for the season JJ, which suggests that the processes influencing the AS Niño 3-AIR relationship are different from those influencing the relationship in the JJ season.

A comparison of Figures 7b and 9 reveals that the weakest AS Niño 3-AIR correlation coincides with the greatest AS Niño 3 waveform skewness. Moreover, the AS relationship is seen to weaken when the Niño 3 waveform skewness increases after the 1970s but strengthens when the Niño 3 waveform skewness declines around the 1990s. These results support the idea that Niño 3 waveform skewness could be related to the weakening AS Niño 3-AIR relationship. Similar arguments hold for the Niño 1+2-AIR relationship.

## 4.3 ENSO waveform skewness and the ENSO-Indian Monsoon Relationship





To gain confidence that the sliding time series shown in Figures 7 and 9 are related, the sliding Niño 3 and Niño 1+2 waveform skewness time series were correlated with the ENSO-AIR sliding correlation time series. The sliding seasonal waveform time series were obtained from the raw JJ and AS waveform time series associated with the Niño 3 and Niño 1+2 indices (Figure S1). In general, the seasonal

waveform skewness time series (Figure S2) were found to be like the ones associated with the full ENSO time series but the seasonal waveform time series were easier to compare with the sliding correlation time series computed for the JJ and AS seasons. For comparison, the AS (JJ) sliding correlation time series were correlated with the seasonal skewness time series computed from the AS (JJ) Niño 3 and Niño 1+2 time series (not shown). The correlation between the sliding skewness and sliding correlation time series were

calculated using different windows lengths. Statistical significance was assessed using Monte Carlo methods (Appendix A).

As shown in Figure 10a, there is a positive correlation between time series for AS Niño 1+2 skewness and waveform skewness and the AS Niño 3-AIR sliding correlation time series. The AS Niño 1+2 skewness relationship with the AS Niño 1+2-AIR sliding correlation time series is statistically

significant at the 5% level for most segment lengths, whereas the corresponding relationship with Niño 1+2 waveform skewness is only statistically at the 5% level for segment lengths ranging from 20 to 25 years. On the other hand, the Niño 3 waveform skewness time series and the Niño 3-AIR sliding correlation time series are correlated with 5% significance for most segment lengths (Figure 10b). As shown in Figure 10c, Niño 3 waveform skewness and skewness are more strongly related to changes in the AS Niño 1+2-AIR

relationship than they are to changes in the AS Niño 3-AIR relationship. The positive correlations seen in Figure 10 support the hypothesis that ENSO nonlinearity is related to changes in the ENSO-AIR relationship (Schulte et al. 2020).

Repeating the analysis for the JJ season revealed weaker relationships between ENSO nonlinearity and the AIR-ENSO relationship. Nevertheless, positive correlations were identified, which agrees with the

idea that time periods of greater ENSO nonlinearity coincide with a weaker ENSO-AIR relationship, as originally suggested by Schulte et al. (2020). Furthermore, the stronger association for the AS season could explain why the Niño 3-AIR relationship weakens after the 1970s while the strength of the JJ relationship is more stable (Figure 9).

### 4.4 ENSO Skewness and ENSO flavors

A possible explanation for the relationships between ENSO nonlinearity and the strength of the ENSO-AIR relationship was determined by compositing AS ENSO indices and AIR anomalies based on





AS ENSO waveform skewness bins (Appendix B). For example, we identified the years for which the AS waveform skewness was greater than the 95[th] and computed the mean AS AIR anomaly for those years.

Figure 11a indicates that the intensity of Niño 3 anomalies is related to Niño 3 waveform skewness. For waveform skewness values less than 0.25, negative Niño 3 indices are preferred, whereas positive

indices are preferred for waveform skewness values greater than 0.25. These results highlight the strong linear relationship between the Niño 3 index and Niño 3 waveform skewness. A similar analysis using the Nino 1+2 index also identified a linear relationship between the Niño 1+2 index and Niño 1+2 waveform skewness, further supporting a relationship between ENSO nonlinearity and ENSO intensity.

Consistent with a linear negative correlation between AIR and the Niño 3 index, AIR anomalies

are preferentially positive for Niño 3 waveform skewness values less than 0 and negative for waveform skewness values greater than 0 (Figure 11b). For the JJ season, the relationship appears to be generally linear, but the relationship between AS Niño 3 waveform skewness and AS AIR anomalies is slightly more complicated. For waveform skewness values ranging from -0.5 to 0.75, the composite mean AIR anomalies rapidly decrease in accord with the tendency for greater Niño 3 indices to be associated with higher

waveform skewness values. However, above a waveform skewness value of 0.75, AIR anomalies no longer decrease with increasing waveform skewness despite the increase in the composite mean Niño 3 indices. This nonlinear relationship also exists between Nino 1+2 waveform skewness and AIR anomalies (not shown) so that our results imply that the linear relationship between ENSO and AIR anomalies degrades for high ENSO waveform skewness, agreeing with the findings from the correlation analysis presented in

Section 4.3.

To diagnose why the Niño 3-AIR relationship breakdowns for high Niño 3 waveform skewness, we composited the magnitude of the TNI index based on Niño 3 waveform skewness. In this analysis, statistical significance of the composite means were assessed relative to the smallest composite mean TNI magnitude (Appendix B).

The results shown in Figure 11c indicate that there is a propensity for the TNI magnitude to increase with increasing Niño 3 waveform skewness. For both seasons, the magnitude of the TNI appears to be preferentially small for Niño 3 waveform skewness ranging from -0.75 to 0. The JJ TNI magnitude tends to be large ($> 0.7$) for Niño 3 waveform skewness above 0.25, and the AS TNI magnitude tends to be greater than 0.7 for waveform skewness values greater than 0.5. Interestingly, we did not find such a

relationship between the magnitude of Niño 3 anomalies and the TNI magnitude (not shown) in AS, suggesting that Niño 3 waveform skewness provides new information about the changing ENSO-Indian monsoon relationship not contained in the actual Niño 3 index.




A similar analysis conducted between Nino 1+2 waveform skewness and TNI magnitude did not reveal any relationship between them (not shown). However, the composite analysis assumes a simultaneous relationship between waveform skewness and the TNI magnitude that seems unlikely to exist given that most skewed Niño 1+2 events coincide with the strongest canonical El Niño events (Figure 5a).

Thus, we correlated sliding ENSO waveform skewness and skewness time series with sliding TNI magnitude time series to determine if time periods of high ENSO nonlinearity coincide with time periods of high TNI magnitude.

As shown in Figure 12, there is a statistically significant relationship between the waveform skewness of the Niño 1+2 and Niño 3 indices and the intensity of the TNI. The results indicate that time

periods of enhanced ENSO nonlinearity occur with time periods of more intense TNI events. A similar relationship also exists between ENSO skewness and TNI magnitude (Figure 13), though the relationship between Niño 3 skewness and TNI magnitude is not statistically significant at the 10% level.

## 5. Discussion/Conclusion

A new waveform skewness index was developed based on principles from nonlinear time series

analysis. The waveform skewness allowed the waveform skewness of individual time series events to be quantified. Using statistical significance tests and waveform spectra, waveform skewness distinguishable from background noise could be identified. Practical applications of the waveform skewness index to ENSO time series highlighted its importance to geophysics. The practical applications led to the identification of numerous positively skewed ENSO events that generally coincide with the strongest El Niño events on

record. The analysis also revealed that ENSO cycles between periods of high and low waveform skewness, with the 1950s being a time period when waveform skewness was relatively low. In contrast, after the 1970s, the waveform skewness of ENSO increased dramatically to a maximum around the 1990s. The fluctuation of waveform skewness is generally consistent with prior work identifying interdecadal changes in ENSO skewness and a prominent regime shift in the 1970s (An, 2009).

We found that Niño 3 waveform skewness is related to the Niño 3-AIR and Niño 1+2-AIR relationships, especially during the AS season. The covariation is such that when the Niño 3 index is strongly and positively skewed, the correlation between the Niño 3 index and AIR anomalies is weaker than the correlation during low waveform skewness time periods. Although greater Niño 3 waveform skewness is positively and linearly correlated with the Niño 3 index, a composite analysis reveals that the

relationship between Niño 3 waveform skewness and AIR anomalies is nonlinear. This nonlinear relationship is such that composite mean AIR anomalies do not intensify as Niño 3 waveform skewness





increases from a moderately high value to a very high value, implying a breakdown of the Niño 3-AIR relationship because the Niño 3 increases with increasing waveform skewness.

A possible explanation for the breakdown of the Niño 3 -AIR relationship during time periods of high Niño 3 waveform skewness is the presence of central equatorial Pacific El Niño events during high
waveform skewness time periods. More specifically, the largest TNI values tend to occur during time periods of high ENSO skewness and waveform skewness. We interpret these findings as a failure of the Nino 1+2 and Nino 3 indices to distinguish different ENSO flavors. For example, the Nino 1+ 2 index can be negative during canonical La Nina events and positive TNI events. As a result, two similar Nino 1+2 indices can be associated with a monsoon surplus promoting La Nina event and a drought producing
positive TNI phase or CP El Nino event (Kumar et al. 2006). As a result of the competing influences of the ENSO flavors, the relationship between ENSO and AIR anomalies weakens. According to our results, the mechanism is stronger during time periods of high ENSO nonidentity because greater ENSO nonlinearity is associated with more intense TNI events that more strongly influence the relationship between standard ENSO indices and AIR.

Our findings support the hypothesis proposed by Schulte et al. (2020) that states that the ENSO-AIR relationship weakens during time periods of high ENSO nonlinearity because the skewness of AIR anomalies is weakly correlated with ENSO skewness. However, our results indicate that the association between ENSO waveform skewness and the ENSO-AIR relationship mainly exists during the AS season. Nevertheless, our findings together those from Schulte et al. (2020) support the idea that the occurrence of
ENSO flavors are related to the nonlinearity of ENSO.

While some studies suggest that changes in the ENSO-Indian monsoon relationship are related to statistical undersampling and stochastic processes (Gershunov et al., 2001; Cash et al., 2017; Yun and Timmermann, 2018), we found robust evidence that changes in the Niño 3 skewness-rainfall anomaly and Niño 3 peak intensity-rainfall anomaly relationships are related to Niño 3 waveform skewness. Thus,
changes in the ENSO-Indian monsoon teleconnection may be predictable to the extent that ENSO flavors can be forecast. Our results agree with previous work showing a link between ENSO flavors and the Indian monsoon (Kumar et al., 1999; Fan et al., 2017) and suggest that ENSO-based forecast should not only include ENSO intensity information, but also information about ENSO flavor. However, our results indicate that the ENSO flavor information may only be useful to Indian monsoon prediction during August and
September.

Although the study focused on ENSO waveform skewness, the generality of the waveform skewness index means that it can be applied to arbitrary time series. For example, the index could be readily



applied to other nonlinear geophysical time series such as the QBO and solar cycle, possibly improving the current understanding of the physics driving the nonlinearity of those time series. Furthermore, because the waveform skewness index is derived using an event decomposition approach, lag composites could be made to identify physical processes associated with the development and decay of nonlinear events. Another

5      application of the waveform skewness index could be in model evaluation, because assessing the ability of a numerical model to capture time series waveform skewness could provide insight into model deficiencies. The waveform skewness index provides new future directions for research focused on understanding nonlinear climate phenomena, numerical model performance, and nonlinear time series in general.





## Appendix A

The statistical significance of the correlation coefficients was estimated using Monte Carlo methods as follows: Firstly, we generated 1000 pairs of red-noise process realizations such that the first member of the pair had lag-1 auto-correlation coefficient equal to the full (non-averaged) input ENSO time series (e.g. Niño 3 index) and the

second member had lag-1 auto-correlation coefficient equal to that of the full input AIR time series. Thus, the first (second) members of all pairs were considered fictitious ENSO (AIR) time series. The lengths of the realizations were also equal to that of the full input AIR and ENSO time series. In the second step, the waveform skewness time series associated with the fictitious ENSO time series were computed and converted into seasonally averaged fictitious ENSO waveform skewness time series. These fictitious seasonally averaged waveform skewness time series were

constructed by identifying the first 12 data points of the full fictitious ENSO time series as the data for the first year, the next 12 points as the data for the second year, and so on. Naturally, the 6th, 7th, 8th ,and 9th data points of the fictitious years were the data for June, July, August and September, respectively. Fictitious seasonally averaged ENSO and AIR time series were created by applying the same averaging approach to all fictitious ENSO and AIR time series.

After creating all the fictious time series, the sliding correlation between the fictitious seasonally averaged

ENSO and AIR time series was then computed for each pair of realizations. Finally, the fictitious seasonally averaged sliding ENSO waveform skewness and sliding correlation time series were correlated for each pair, resulting in a null distribution of correlation coefficients. The absolute value of the correlation coefficients was computed and the 90th percentile of the resulting distribution was calculated to estimate the critical level of the test corresponding to the 10% significance level. The Monte Carlo method was implemented for each segment length separately because the critical

level of the test was found to depend on the chosen segment length.



**Appendix B**

Composite analyses were used to determine the relationship between Niño 3 waveform skewness and anomalies for AIR and SST. The composite analysis was conducted by first creating Niño 3 waveform skewness bins by calculating percentiles of a distribution comprising the waveform skewness values associated with the seasonally

averaged (JJ or AS) Niño 3 waveform skewness. The lower and upper bounds of the first bins were the minimum waveform skewness value and the 20th percentile of the distribution, respectively. similarly, the second bin had lower and upper bounds equal to the 5th and 25 percentiles of the waveform skewness value distribution. More generally, for n > 1, the n-th bin had lower and upper bounds equal to the $(5n-5)$-th and $(5n+15)$-th percentiles of the waveform skewness value distribution. After creating the bins, all seasonally averaged Niño 3 and AIR anomalies

associated with Niño 3 waveform skewness values falling into a bin were averaged, allowing us to determine how AIR and Niño 3 index anomalies are related to Niño 3 waveform skewness. The statistical significance of the composite means was evaluated using a two-sample t test applied at the 10% significance level. The null hypothesis in this case was that mean was equal to zero. The composite analysis was performed using AIR, Niño 3 anomalies, and Niño 3 waveform skewness associated with the JJ and AS seasons separately so that a composite mean AIR

anomaly for the AS (JJ) season represented the average AS (JJ) AIR anomaly expected for a specific AS (JJ) Niño 3 waveform skewness range.

Another composite analysis was also conducted in which the magnitude of the JJ or AS TNI was composited based on its associated JJ or AS Niño 3 waveform skewness. In this case, the statistical significance was assessed using the null hypothesis that the composite mean is equal to the smallest computed composite mean for the season

in question. Thus, we determined if the composite mean for bins with larger TNI magnitudes are significantly different from the bin with smallest TNI magnitude.



**Conflicts of Interest.** The authors declare that is no conflicts of interest.



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



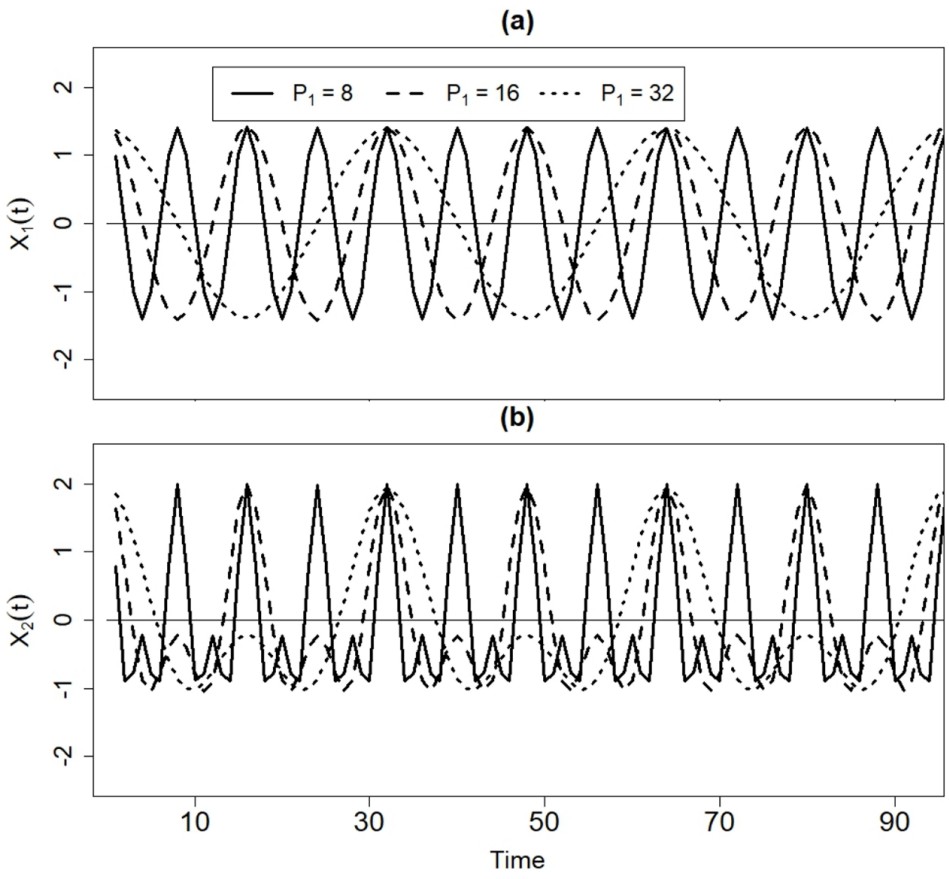

Figure 1. (a) The time series $X_1(t)$ and (b) $X_2(t)$ for different values of $P_1$.

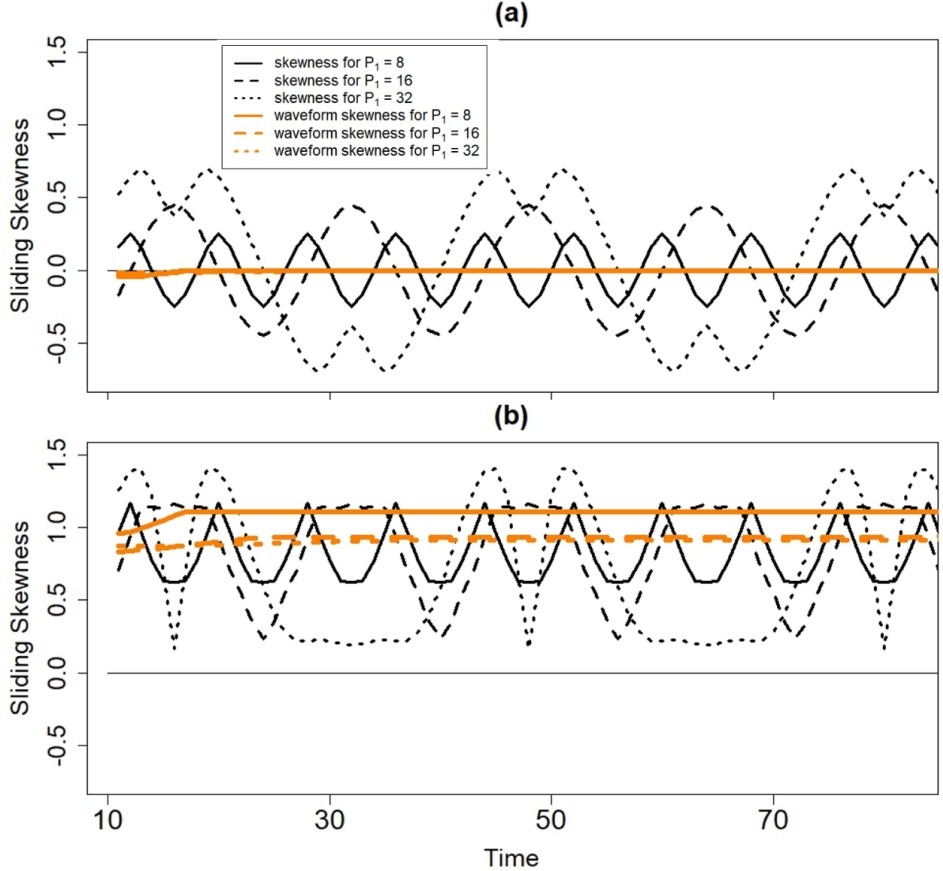

Figure 2. Sliding skewness and waveform time series associated with (a) $X_1(t)$ and (b) $X_2(t)$ for different values of $P_1$. The segment lengths used to create the sliding skewness and waveform skewness time series were equal to 20 data points.





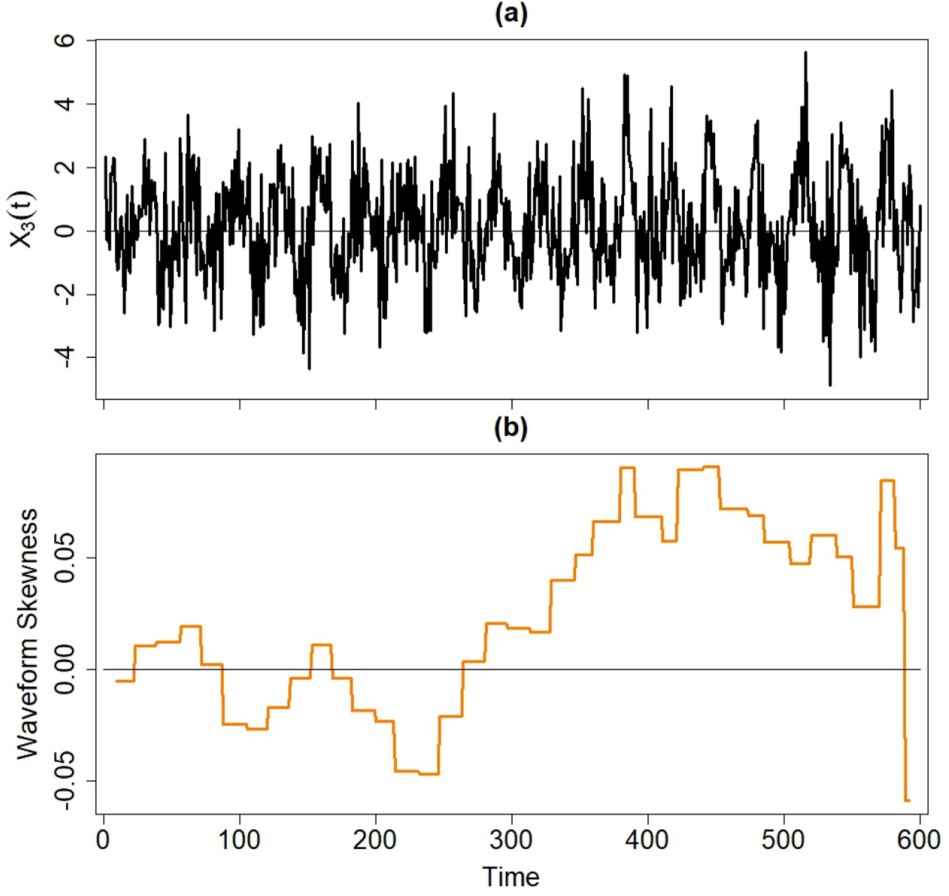

Figure 3. (a) The time series $X_3(t)$ and (b) its waveform skewness time series associated with the point (32,32) in the waveform spectrum.



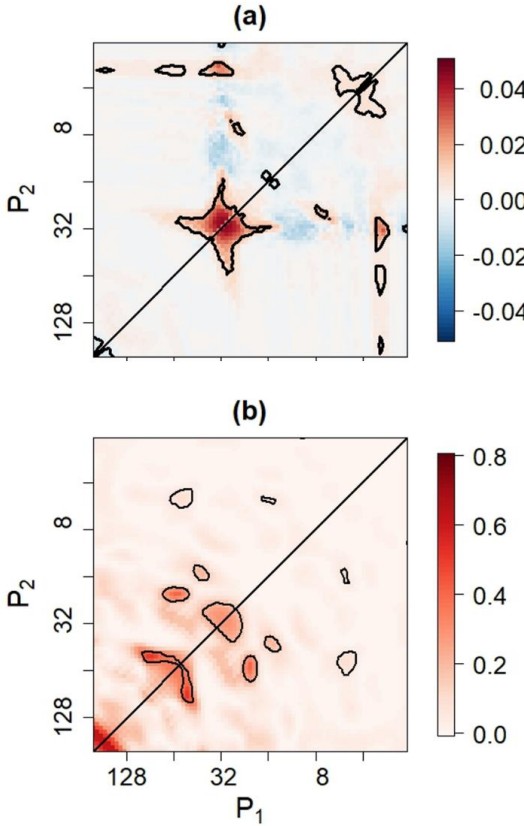

Figure 4. (a) The waveform skewness and (b) auto-bicoherence spectra associated with $X_3(t)$. Contours enclose regions of 5% statistical significance.



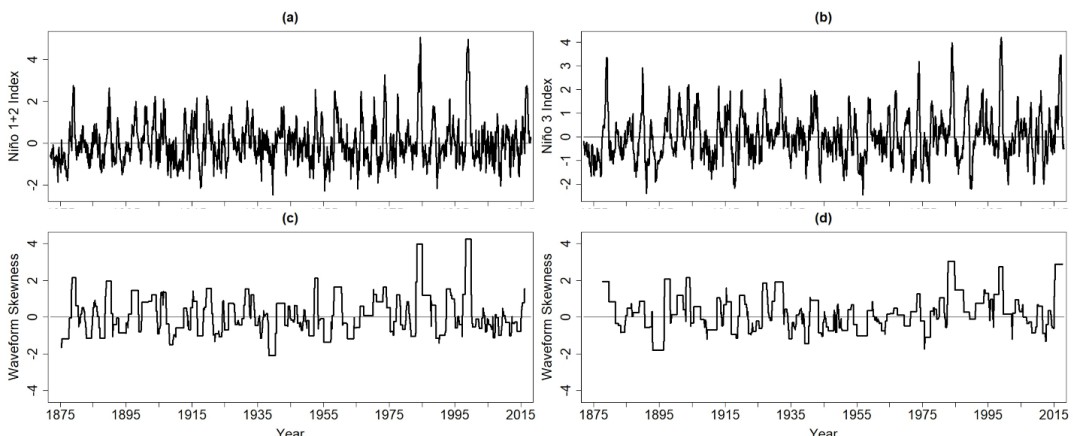

Figure 5. The standardized (a) Niño 1+2 and (b) Niño 3 indices together with the waveform skewness time series of the (c) Niño 1+2 and (d) Niño 3 indices.



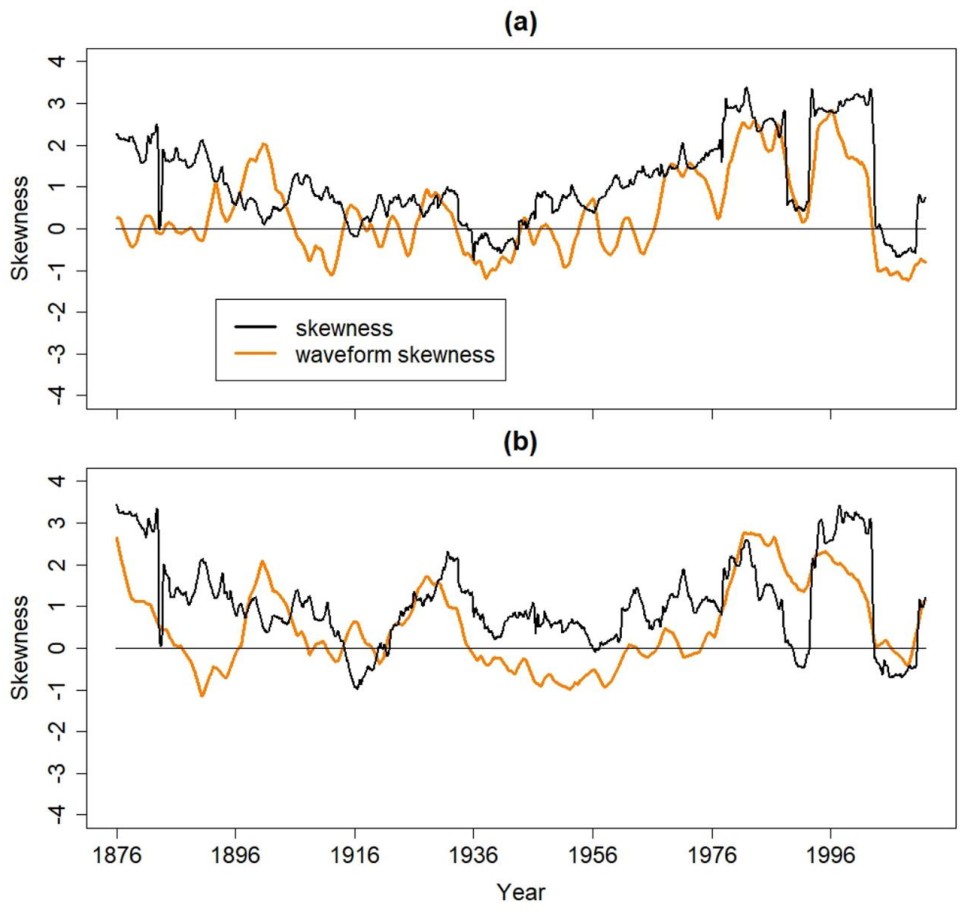

Figure 6. 10-year sliding skewness and waveform skewness times series corresponding to the (a) Niño 1+2 and (b) Niño 3 indices.





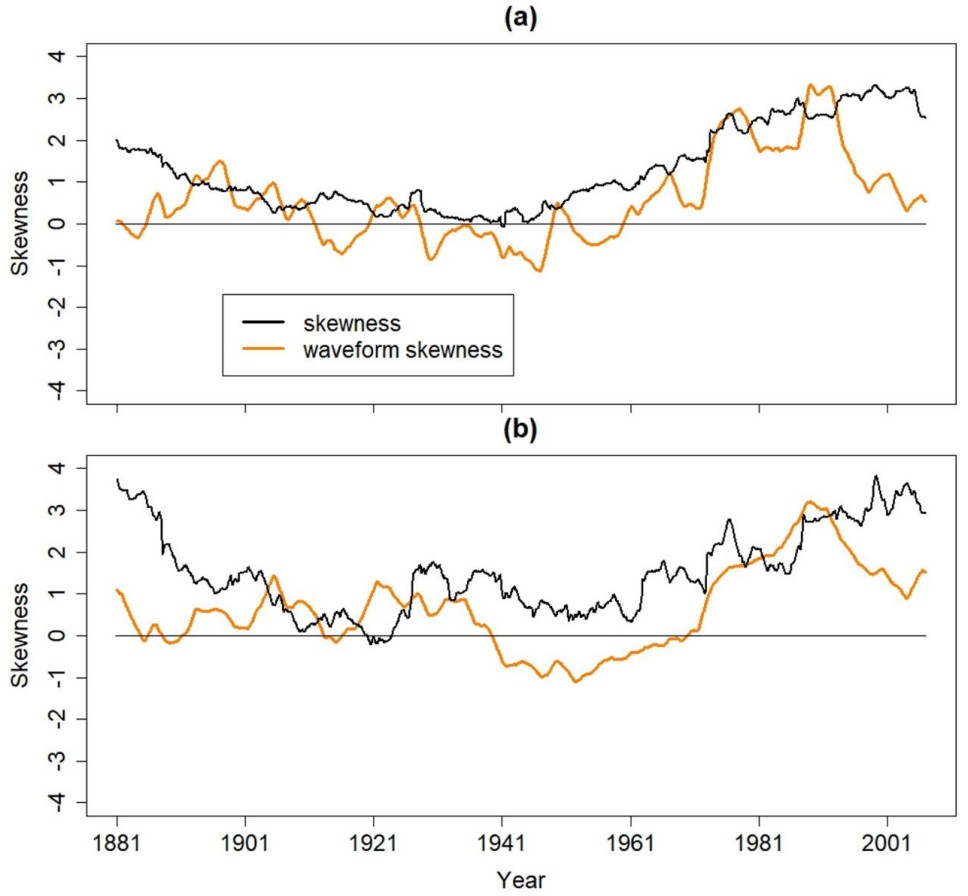

Figure 7. 20-year sliding skewness and waveform skewness times series corresponding to the (a) Niño 1+2 and (b) Niño 3 indices.



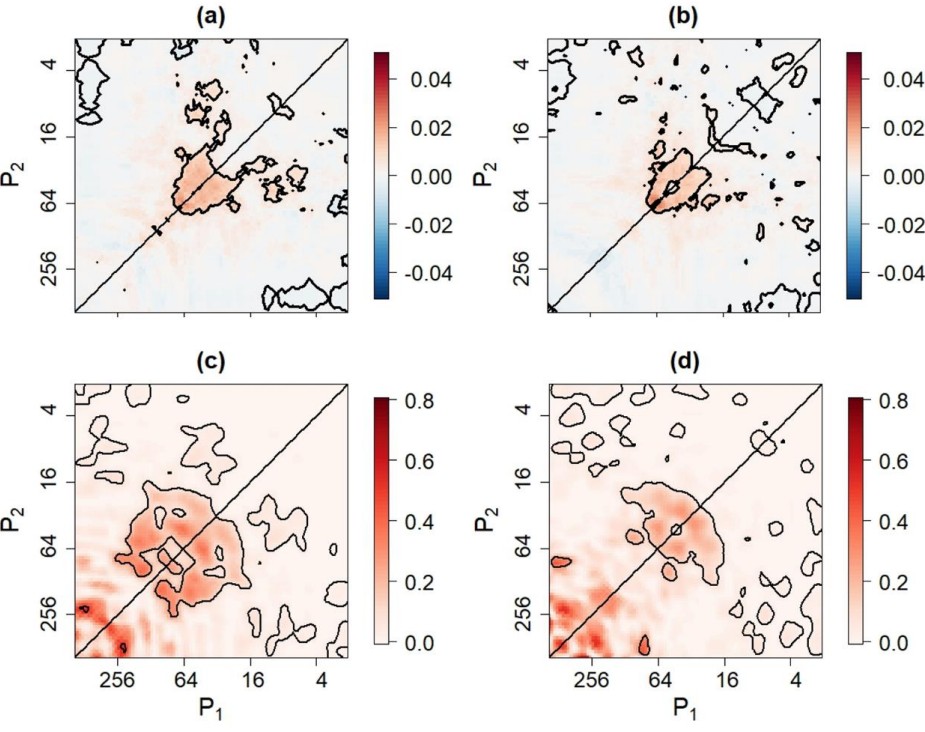

Figure 8. The waveform skewness spectra corresponding to the Niño 1+2 and Niño 3 indices. The corresponding auto-bicoherence spectra of the (c) Niño 1+2 and (d) Niño 3 indices. Contours enclose regions of 5% statistical significance.





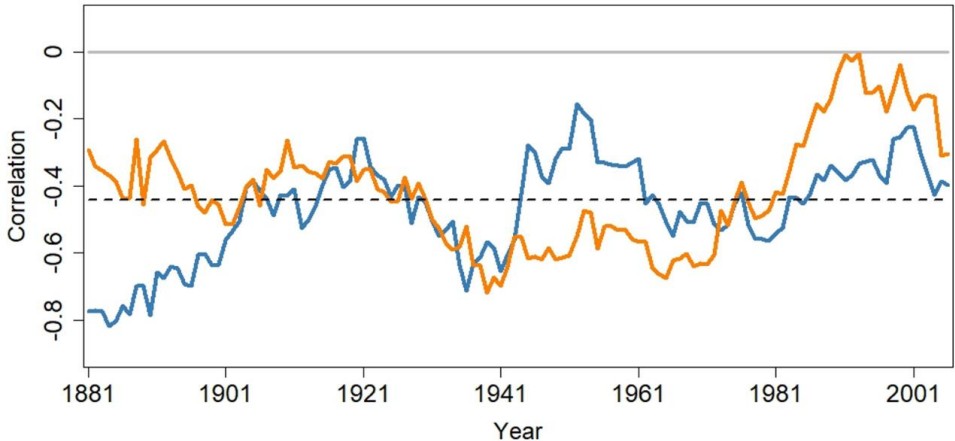

Figure 9. Sliding correlation between the Niño 3 index and the AIR anomalies for the JJ and AS seasons. The horizonal
dashed line represents the 5% significance bound.


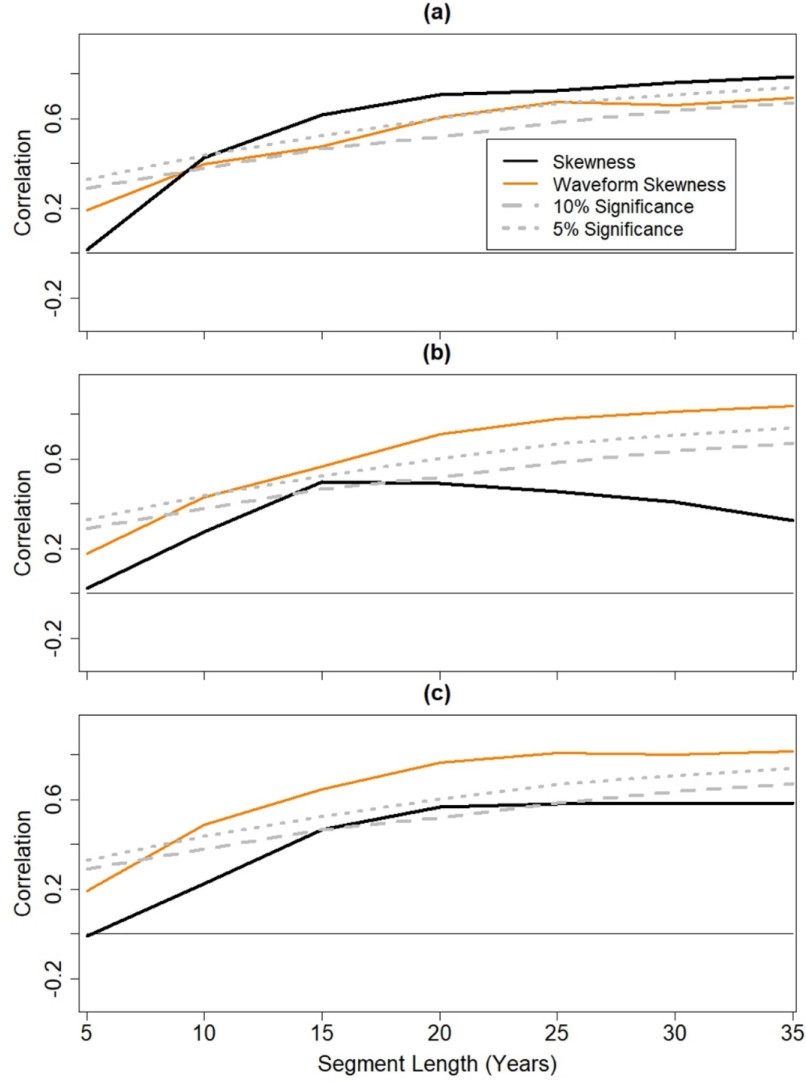

Figure 10. Correlation between time series for Niño 1+2 skewness and waveform skewness and the sliding correlation time series calculated between the Niño 1+2 index and AIR anomalies. (b) Correlation between time series for Niño 3 skewness and waveform skewness and the sliding correlation time series calculated between the Niño 3 index and AIR anomalies. (c) Correlation between time series for Niño 3 skewness and waveform skewness and the sliding correlation time series calculated between the Niño 1+2 index and AIR anomalies.





Figure 11. (a) Composite mean (a) Niño 3 index, AIR anomaly, and (b) TNI magnitude as a function of Niño 3 waveform skewness. Dots indicate statistical significance at the 5% level.



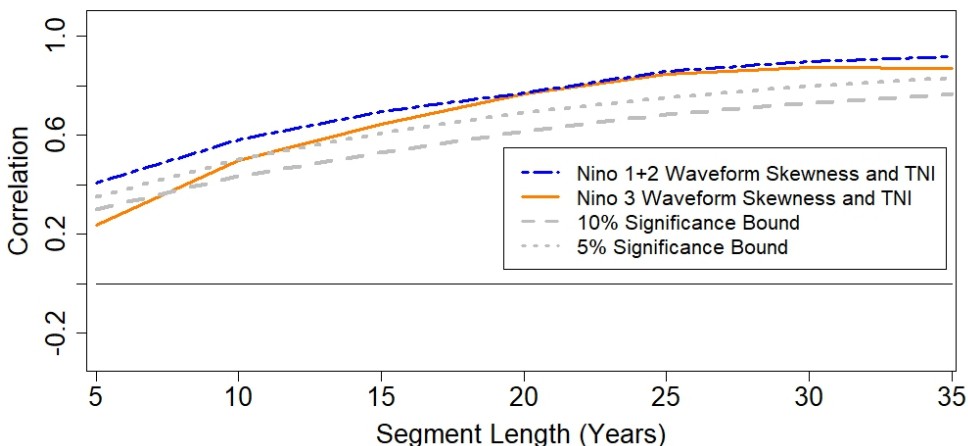

Figure 12. Correlation between waveform skewness and TNI magnitude.





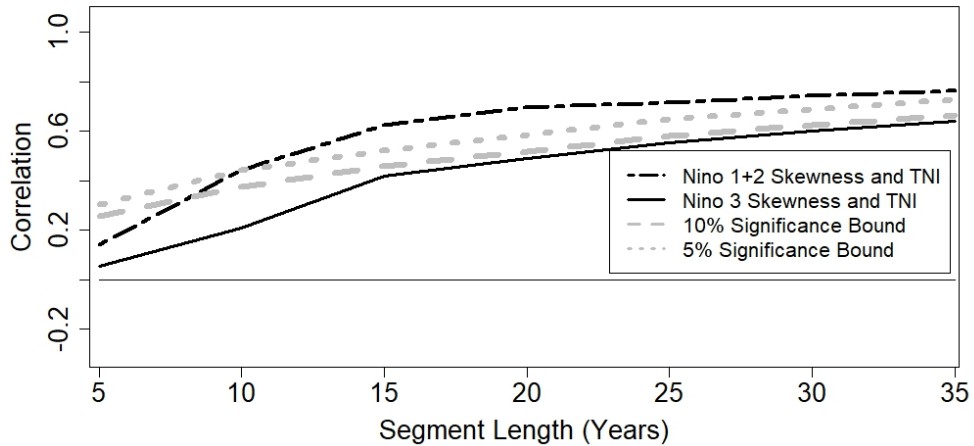

Figure 13. Correlation between skewness and TNI magnitude.