# Peer review of "A Waveform Skewness Index for Measuring Time Series Nonlinearity and its Applications to the ENSO-Indian Monsoon Relationship"

_Nonlinear Processes in Geophysics, 2020_

## Short Comment (SC1) · 19 Dec 2020

FYI, showed that skewness results when solving the wave equations along the equator, e.g. for ENSO. This generates a nonlinear amplification that can differ with respect to the sign of the amplitude excursion.

Pukite, P., Coyne, D., & Challou, D. (2019). Mathematical Geoenergy: Discovery, Depletion, and Renewal (Vol. 241). John Wiley & Sons. https://agupubs.onlinelibrary.wiley.com/doi/10.1002/9781119434351.ch12

2020-48, 2020.

For a fluid sheet of average thickness $D$, the vertical tidal elevation $\zeta$, and the horizontal velocity components $u$ and $v$ (in the latitude $\varphi$ and longitude $\lambda$ directions), this is the set of Laplace's tidal equations:

$$\frac{\partial \zeta}{\partial t} + \frac{1}{a\cos(\varphi)}\left[\frac{\partial}{\partial \lambda}(uD) + \frac{\partial}{\partial \varphi}(vD\cos(\varphi))\right] = 0,$$

$$\frac{\partial u}{\partial t} - v(2\Omega\sin(\varphi)) + \frac{1}{a\cos(\varphi)}\frac{\partial}{\partial \lambda}(g\zeta + U) = 0, \quad (11.1)$$

$$\frac{\partial v}{\partial t} + u(2\Omega\sin(\varphi)) + \frac{1}{a}\frac{\partial}{\partial \varphi}(g\zeta + U) = 0,$$

where $\Omega$ is the angular frequency of the planet's rotation, $g$ is the planet's gravitational acceleration at the mean ocean surface, $a$ is the planetary radius, and $U$ is the external gravitational tidal forcing potential. The goal is that along the equator, for $\varphi$ at zero, we can reduce these three equations into one.

As we will re-derive a simplification of these equations in the next chapter when we discuss ENSO, it is enough at the present to point to a simplifying differential relation below:

$$\frac{\partial \zeta}{\partial \varphi} = \frac{\partial \zeta}{\partial t}\frac{\partial t}{\partial \varphi} \quad (11.2)$$

Via this and the other simplifying assumption of the Coriolis forces canceling at the equator, we obtain the following potentially highly nonlinear result:

$$\zeta(t) = \sin\left(\sqrt{A}\sum_{i=1}^{i=N}k_i\sin(\omega_i t) + \theta_i\right) \quad (11.3)$$

topological insulators as applied to equatorial phenomena such as QBO and ENSO will give rise to Rossby, Kelvin, and Yanai waves (Delplace et al., 2017).

Now consider that the QBO itself is precisely the $\partial v/\partial t$ term (the horizontal longitudinal acceleration of the fluid, i.e., leading to the observed characteristic waveform) which can be derived from the above by applying the solution to Laplace's third tidal equation in simplified form:

$$\frac{\partial v}{\partial t} = \cos\left(\sqrt{A}\sum_{i=0}^{i=N}k_i\sin(\omega_i t) + \theta_i\right) \quad (11.5)$$

Note again that this is the QBO acceleration and not the QBO velocity, which is usually reported.

**11.1.1. Harmonics**

The form of the last equation suppresses large amplitudes, thus leading to the formation of harmonics of the fundamental frequencies $\omega_i$. One potential candidate for $\omega_i$ is to apply a strong annual pulse to one of the known lunar monthly periods, such as the lunar or draconic cycle of 27.212 days. This will generate the physically aliased periods shown in the Table 11.1.

Note that the period 2.363 years corresponding to the fourth entry corresponds to approximately 28 months.

Figure 11.1 shows a multiple linear regression fit of the terms from the above table using a very short training interval, showing good cross-validation to other parts of the periodic QBO waveform. Thus, the naturally resonant or chaotic solution is secondary to this larger forced response (Osipov et al., 2007; Wang et al., 2013).

**Fig. 1.** Solution approach

[Figure]

increasing LTE modulation $\rightarrow$

**Fig. 2.** Skewness results from asymmetric folding of the nonlinear mapping

---

## Referee Comment (RC1) · Anonymous Referee #1 · 11 Jan 2021

The present manuscript presents an accurate analysis of the use of the Waveform Skewness Index, as compared to the traditional (sliding) skewness coefficient and its dependence as a function of the sliding period, presenting possible drawbacks of this skewness estimator. Authors also study the robustness of the link between El-Niño skewness and AIR and also between TNI and AIR using the proposed Waveform Skewness Index. Significance tests of those links (measured by correlations) are assessed by Monte-Carlo experiments. The manuscript is very well written and complements very well previous recent works of the authors, presenting original material being thus suitable for NPG. It is believed however, that the manuscript can be further improved after considering a few points of reflection.

1) Pg. 2, lines 3-4. Skewness of a time-series is not an exclusive feature of non-linear systems as authors implicitly supposed to. As an example, we can simply consider a stochastic linear process driven by a non-Gaussian skewed noise where the noise skewness comes as a function of the signal skewness and of the signal autocovariance function (e.g. Pires and Hannachi 2021 (PH21 in the sequel))

2) Pg. 3 In the paragraph (line 8-13) it is opportune to mention the test of nonlinearity of the El-Niño, followed in PH21. It relies on the standardized difference between the El-Niño bispectrum and the bispectrum of a linear non-Gaussian process fitting El-Niño reproducing the same spectrum and skewness.

3) pg. 4, line 4 Add a reference for trans-Niño index (e.g. Trenberth and Stepaniak 2001).

4) pg. 4 Line 7 '...implicated has an SST pattern' must be '...implicated as an SST pattern' ?

5) Pg.5, Eq.4 divide by s**3.

6) Pg. 5 line 17. Say in text that the tested length N in the computation of the sliding skewness was 20 months to facilitate its comparison with the periods 8,16 and 32. It seems that the effect of oscillations on the skewness estimation appears to be significant only when the largest Fourier period in the time-series is of the order of N or some when some long-term trend id present. Please comment that aspect.

7) Pg 7, line 13 siding → sliding

8) Pg. 7, Sec. 3.3. The Waveform skewness timeseries is a modified timeseries keeping some characteristics of the original raw timeseries. What in effect is kept? For instance, how much is correlation and coherency spectrum? Could you develop some considerations about this issue.

9) Pg. 11 Lines 15-18. You should stress that similar El-Niño auto-bicoherence with identical triads of periods was obtained by Pires and Hannachi (2021) in its Fig. 9a.

10) Fig. 9 Indicate the meaning of line colors (JJ and AS seasons).

11) Pg. 9. The series x3(t) given by Eq. 11,12 is very interesting to test the phase synchronization and the time-varying skewness, through a quadratic growing (not stationary) amplitude of the P3 component. This kind of model raises the idea of another model that it will be very worth to test. In fact, skewness may not uniquely come from phase synchronization but also from a correlation between the amplitude gamma(t) and the phase phi3(t) which should oscillate instead of being taken constant. The bispectrum sum over bi-frequencies equals the skewness. However, the bispectrum can be decomposed into a phase synchronization term and the above referred correlation term, thus putting in evidence the two possible mechanisms of generating skewness. For details see Sec. 5.1.1 of Pires and Hannachi (2021), in particular the decomposition in its Eq. 12.

References:

Carlos A. L. Pires & Abdel Hannachi (2021) Bispectral analysis of nonlinear interaction, predictability and stochastic modelling with application to ENSO, Tellus A: Dynamic Meteorology and Oceanography, 73:1, 1-30, DOI: 10.1080/16000870.2020.1866393

https://www.tandfonline.com/action/showCitFormats?doi=10.1080%2F16000870.2020.1866393

Trenberth, K. E., & Stepaniak, D. P. (2001). Indices of El Niño Evolution. Journal of Climate, 14(8), 1697–1701. doi:10.1175/1520-0442(2001)014<1697:lioeno>2.0.co;2

---

## Referee Comment (RC2) · RATHINASAMY MAHESWARAN (Referee) · 17 Feb 2021

Review of the Manuscript "A Waveform Skewness Index for Measuring Time Series Nonlinearity and its Applications to the ENSO-Indian Monsoon Relationship" by Justin Schulte et al.

The authors propose a new measure for nonlinearity using waveform index.The paper is well written and informative. Indeed, the paper is of great interest to NPG readers. But before acceptance, the authors should provide valid responses for the following comments. 1. There are several other nonlinearity measures available, and the author can use them to compare the results. 2. Page 2, lines1-2, there could several other

manifestations of the nonlinearity in a time series, may measure such as transfer entropy could capture them. 3. Page 8 lines 1-2, it is not clear, please elaborate 4. Fig 6 and 7, state the reasons why the authors have chosen 10 and 20 year sliding window. 5. The authors have used correlation and sliding correlation to measure the relationship between the nonlinearity in ENSO and AIR anamolies, they could have very well used other measures such as mutual information. That would be more appropriate. 6. The author have chosen AIR for the analysis, many studies have shown that the relationship between ENSO and indian rainfall is spatially variable, in that context how the application of AIR is justified and the results are meaningful.

Other minor comments, The author refer to the paper Schulte 2020 many times in the introduction, but I could not find it in the references Please correct the Eq.4 denominator

---

## Author Comment (AC1) · 30 Mar 2021

The present manuscript presents an accurate analysis of the use of the Waveform Skewness Index, as compared to the traditional (sliding) skewness coefficient and its dependence as a function of the sliding period, presenting possible drawbacks of this skewness estimator. Authors also study the robustness of the link between El-Niño skewness and AIR and also between TNI and AIR using the proposed Waveform Skewness Index. Significance tests of those links (measured by correlations) are assessed by Monte-Carlo experiments. The manuscript is very well written and complements very well previous recent works of the authors, presenting original material being thus suitable for NPG. It is believed however, that the manuscript can be further improved after considering a few points of reflection.

1) Pg. 2, lines 3-4. Skewness of a time-series is not an exclusive feature of non-linear systems as authors implicitly supposed to. As an example, we can simply consider a stochastic linear process driven by a non-Gaussian skewed noise where the noise skewness comes as a function of the signal skewness and of the signal autocovariance function (e.g. Pires and Hannachi 2021 (PH21 in the sequel))

The authors agree that we need to clarify that waveform skewness could arise from other processes besides non-linear ones. As such, a few sentences will be added in the methods section to comment about other sources of waveform skewness.

2) Pg. 3 In the paragraph (line 8-13) it is opportune to mention the test of nonlinearity of the El-Niño, followed in PH21. It relies on the standardized difference between the El-Niño bispectrum and the bispectrum of a linear non-Gaussian process fitting El-Niño reproducing the same spectrum and skewness.

The authors appreciate the referral to that test. It will now be mentioned on Page 3.

**3) pg. 4, line 4 Add a reference for trans-Niño index (e.g. Trenberth and Stepaniak 2001).**

The missing reference will be added to the revised manuscript.

**4) pg. 4 Line 7 '. . .implicated has an SST pattern' must be '. . .implicated as an SST pattern' ?**

Thank you for pointing out the typographic error. It will be corrected in the revised manuscript.

**5) Pg.5, Eq.4 divide by s\*\*3.**

Thank you for pointing out the typographic error. It will be corrected in the revised manuscript.

**6) Pg. 5 line 17. Say in text that the tested length N in the computation of the sliding skewness was 20 months to facilitate its comparison with the periods 8,16 and 32. It seems that the effect of oscillations on the skewness estimation appears to be significant only when the largest Fourier period in the timeseries is of the order of N or some when some long-term trend id present. Please comment that aspect.**

The authors agree that the effect of the oscillations is only significant when the period of the oscillation is roughly greater than or equal the chosen sliding segment length. We will add a few sentences in the revised manuscript to reflect this observation, as it could help researchers decide an appropriate segment length to use their analyses.

**7) Pg 7, line 13 siding $\rightarrow$ sliding**

Thank you for pointing out the typographic error. It will be corrected in the revised manuscript.

**8) Pg. 7, Sec. 3.3. The Waveform skewness timeseries is a modified timeseries keeping some characteristics of the original raw timeseries. What in effect is kept? For instance, how much is correlation and coherency spectrum? Could you develop some considerations about this issue.**

The authors agree that it would be important understand what components of the original time series remain after the transformation to waveform skewness. This topic will be explored in the revised manuscript by correlating the transformed time series with the original one. A few sentences will be added to inform the readers about how much the transformed time series differs from the original time series.

**9) Pg. 11 Lines 15-18. You should stress that similar El-Niño auto-bicoherence with identical triads of periods was obtained by Pires and Hannachi (2021) in its Fig. 9a.**

The authors appreciate the referral to the Pires and Hannachi (2021) paper. We will add a discussion about how our results compare to those of Pires and Hannachi (2021) in the revised manuscript.

**10) Fig. 9 Indicate the meaning of line colors (JJ and AS seasons).**

Color scheme will be described in the revised manuscript.

11) Pg. 9. The series x3(t) given by Eq. 11,12 is very interesting to test the phase synchronization and the time-varying skewness, through a quadratic growing (not stationary) amplitude of the P3 component. This kind of model raises the idea of another model that it will be very worth to test. In fact, skewness may not uniquely come from phase synchronization but also from a correlation between the amplitude gamma(t) and the phase phi3(t) which should oscillate instead of being taken constant. The bispectrum sum over bi-frequencies equals the skewness. However, the bispectrum can be decomposed into a phase synchronization term and the above referred correlation term, thus putting in evidence the two possible mechanisms of generating skewness. For details see Sec. 5.1.1 of Pires and Hannachi (2021), in particular the decomposition in its Eq. 12.

The authors thank the reviewer for pointing out the interesting relationship between a bi-spectrum and the covariance between amplitude and phase. The authors agree that it is an important topic, but the authors are concerned about including additional experimental tests in the manuscript because the manuscript already includes 13 Figures. Given this concern, we will instead include an example in the supplementary material that highlights how correlaton between phase and amplitude can crate waveform skewness.

**References**

Carlos A. L. Pires & Abdel Hannachi (2021) Bispectral analysis of nonlinear interaction, predictability and stochastic modelling with application to ENSO, Tellus A: Dynamic Meteorology and Oceanography, 73:1, 1-30, DOI: 10.1080/16000870.2020.1866393

---

## Author Comment (AC2) · 30 Mar 2021

**maheswaran27@yahoo.co.in**

**Review of the Manuscript "A Waveform Skewness Index for Measuring Time Series**

**Nonlinearity and its Applications to the ENSO-Indian Monsoon Relationship" by Justin SchulteSchulte et al.**

**The authors propose a new measure for nonlinearity using waveform index.The paper is well written and informative. Indeed, the paper is of great interest to NPG readers. But before acceptance, the authors should provide valid responses for the following comments. There are several other nonlinearity measures available, and the author can use them to compare the results. Page 2, lines1-2, there could several other manifestations of the nonlinearity in a time series, may measure such as transfer entropy could capture them.**

The reviewer offers several useful and appropriate suggestions for refining the methods used to identify associations between ENSO and Indian rainfall. We quite agree with the spirit of these comments. In this case, however, we specifically intend for this paper to respond to an active debate in the literature regarding the randomness vs. non-randomness of shifting ENSO-Monsoon relationships. As this literature has generally used linear correlation methods to diagnose associations and has considered generalized indicators of monsoon strength like All-India Rainfall (AIR), we feel it is important to hew to these established approaches wherever possible. We want for this paper to be as accessible as possible for researchers in this field, and for our results to be readily comparable to other studies. In the revised manuscript we will add text to the introduction to make this motivation clear. Specific reviewer recommendations on statistical approaches will also be addressed in the text, as noted in our responses to specific comments below.

**3. Page 8 lines 1-2, it is not clear, please elaborate 4. Fig 6 and 7, state the reasons why the authors have chosen 10 and 20 year sliding window.**

The authors agree that some justification for the selection of the 10-year and 20-year intervals are needed. The main reason why we used the 20-year segments is because that interval length is close to that used in previous works whose focus was on the AIR-ENSO relationship. The 10-year sliding interval was chosen because it is half of the 20-year interval. Although the main conclusions of our results do not change if other interval lengths are chosen, we will note in the revised manuscript that other interval lengths were considered.

**5. The authors have used correlation and sliding correlation to measure the relationship between the nonlinearity in ENSO and AIR anomalies, they could have very well used other measures such as mutual information. That would be more appropriate.**

While the authors agree that there are other ways of measuring the association between two variables, correlation is the most frequent way AIR has been related to ENSO in the debate about the randomness of the AIR-ENSO relationship. As such, the authors feel that using the more common correlation method would allow researchers to more directly compare our results to those of previous works.

**6. The author have chosen AIR for the analysis, many studies have shown that the relationship between ENSO and indian rainfall is spatially variable, in that context how the application of AIR is justified and the results are meaningful.**

Although the authors agree that considering the spatial variability is necessary for all full understanding of the Indian monsoon, the inclusion of a spatial variability analysis is beyond the scope of the paper. This study is focused on the well-studied relationship between AIR and ENSO and contributes to the ongoing debate about the randomness of the AIR-ENSO relationship. However, because the omittance of a spatial variability does represent an important limitation of the study, a dedicated discussion of the limitation will be added to the discussion section of the revised manuscript.

**Other minor comments, The author refer to the paper Schulte 2020 many times in the introduction, but I could not find it in the references Please correct the Eq.4 denominator.**

Thank you for the identifying the reference issue. The reference issue will be corrected in the revised manuscript.

---

## Author Comment (AC3) · 30 Mar 2021

Thank you for your comment. The reference will be added to our reference list and text.

---

## Author Response (AR1)

**The present manuscript presents an accurate analysis of the use of the Waveform Skewness Index, as compared to the traditional (sliding) skewness coefficient and its dependence as a function of the sliding period, presenting possible drawbacks of this skewness estimator. Authors also study the robustness of the link between El-Niño skewness and AIR and also between TNI and AIR using the proposed Waveform Skewness Index. Significance tests of those links (measured by correlations) are assessed by Monte-Carlo experiments. The manuscript is very well written and complements very well previous recent works of the authors, presenting original material being thus suitable for NPG. It is believed however, that the manuscript can be further improved after considering a few points of reflection.**

**1) Pg. 2, lines 3-4. Skewness of a time-series is not an exclusive feature of non-linear systems as authors implicitly supposed to. As an example, we can simply consider a stochastic linear process driven by a non-Gaussian skewed noise where the noise skewness comes as a function of the signal skewness and of the signal autocovariance function (e.g. Pires and Hannachi 2021 (PH21 in the sequel))**

The authors agree that we need to clarify that waveform skewness could arise from other processes besides non-linear ones. As such, a few sentences were added on Page 7 Lines 17 to 20 of the revised manuscript to comment about other sources of waveform skewness.

**2) Pg. 3 In the paragraph (line 8-13) it is opportune to mention the test of nonlinearity of the El-Niño, followed in PH21. It relies on the standardized difference between the El-Niño bispectrum and the bispectrum of a linear non-Gaussian process fitting El-Niño reproducing the same spectrum and skewness.**

The authors appreciate the referral to that test. A brief discussion has been added on Page 3 Lines 13 to 19.

**3) pg. 4, line 4 Add a reference for trans-Niño index (e.g. Trenberth and Stepaniak 2001).**

The missing reference was added to the revised manuscript on Page 4 Line 10.

**4) pg. 4 Line 7 '. . .implicated has an SST pattern' must be '. . .implicated as an SST pattern' ?**

Thank you for pointing out the typographical error. It was corrected on Page 4 Line 12 of the revised manuscript.

**5) Pg.5, Eq.4 divide by s\*\*3.**

Thank you for pointing out the typographic error. It was corrected on Page 5 Line 12 of the revised manuscript.

**6) Pg. 5 line 17. Say in text that the tested length *N* in the computation of the sliding skewness was 20 months to facilitate its comparison with the periods 8,16 and 32. It seems that the effect of oscillations on the skewness estimation appears to be significant only when the largest Fourier period in the time-series is of the order of N or some when some long-term trend id present. Please comment that aspect.**

The authors agree that the effect of the oscillations is only significant when the period of the oscillation is roughly greater than or equal the chosen sliding segment length. We added a few sentences to the revised

manuscript on Page 6 Lines 1-5 to reflect this observation, as it could help researchers decide an appropriate segment length to use their analyses.

**7) Pg 7, line 13 siding → sliding**

Thank you for pointing out the typographic error. It was corrected in the revised manuscript.

**8) Pg. 7, Sec. 3.3. The Waveform skewness timeseries is a modified timeseries keeping some characteristics of the original raw timeseries. What in effect is kept? For instance, how much is correlation and coherency spectrum? Could you develop some considerations about this issue.**

The authors agree that it is important to understand what components of the original time series remain after the transformation to waveform skewness. This topic was explored in the revised manuscript by correlating waveform skewness time series with the corresponding time series. A few sentences were added on Page 7 Lines 11 to 16 of the revised manuscript to inform the readers about how much the transformed time series differs from the original time series.

**9) Pg. 11 Lines 15-18. You should stress that similar El-Niño auto-bicoherence with identical triads of periods was obtained by Pires and Hannachi (2021) in its Fig. 9a.**

The authors appreciate the referral to the Pires and Hannachi (2021) paper. We added a discussion about how our results in relation to those of Pires and Hannachi (2021) in the revised manuscript on Page 12 Lines 6-9.

**10) Fig. 9 Indicate the meaning of line colors (JJ and AS seasons).**

Color scheme is now described in the revised manuscript.

**11) Pg. 9. The series x3(t) given by Eq. 11,12 is very interesting to test the phase synchronization and the time-varying skewness, through a quadratic growing (not stationary) amplitude of the P3 component. This kind of model raises the idea of another model that it will be very worth to test. In fact, skewness may not uniquely come from phase synchronization but also from a correlation between the amplitude gamma(t) and the phase phi3(t) which should oscillate instead of being taken constant. The bispectrum sum over bi-frequencies equals the skewness. However, the bispectrum can be decomposed into a phase synchronization term and the above referred correlation term, thus putting in evidence the two possible mechanisms of generating skewness. For details see Sec. 5.1.1 of Pires and Hannachi (2021), in particular the decomposition in its Eq. 12.**

The authors thank the reviewer for pointing out the interesting relationship between a bi-spectrum and the covariance between amplitude and phase. The authors agree that it is an important topic, but the authors are concerned about including additional experimental tests in the manuscript because the manuscript already includes 13 Figures. Given this concern, we instead included an example in the supplementary material that highlights how the correlaton between phase and amplitude can generate waveform skewness. The reader is referred to the Supplementary material in a brief discussion located on Page 10 Lines 21 to 24 of the revised manuscript.

RATHINASAMY MAHESWARAN (Referee)

maheswaran27@yahoo.co.in

Review of the Manuscript "A Waveform Skewness Index for Measuring Time Series

Nonlinearity and its Applications to the ENSO-Indian Monsoon Relationship" by Justin SchulteSchulte et al.

The authors propose a new measure for nonlinearity using waveform index.The paper is well written and informative. Indeed, the paper is of great interest to NPG readers. But before acceptance, the authors should provide valid responses for the following comments. There are several other nonlinearity measures available, and the author can use them to compare the results. Page 2, lines1-2, there could several other manifestations of the nonlinearity in a time series, may measure such as transfer entropy could capture them.

The reviewer offers several useful and appropriate suggestions for refining the methods used to identify associations between ENSO and Indian rainfall. We quite agree with the spirit of these comments. In this case, however, we specifically intend for this paper to respond to an active debate in the literature regarding the randomness vs. non-randomness of shifting ENSO-Monsoon relationships. As this literature has generally used linear correlation methods to diagnose associations and has considered generalized indicators of monsoon strength like All-India Rainfall (AIR), we feel it is important to hew to these established approaches wherever possible. We want for this paper to be as accessible as possible for researchers in this field, and for our results to be readily comparable to other studies. Neverthess, in the revised manuscript, we added text to the discussion section on Pages 16 and 17 of the revised manuscript to highlight how the exclusive use of linear correlation methods is a limitation of the study..

3. Page 8 lines 1-2, it is not clear, please elaborate 4. Fig 6 and 7, state the reasons why the authors have chosen 10 and 20 year sliding window.

The authors agree that some justification for the selection of the 10-year and 20-year intervals is needed. The main reason why we used the 20-year segments is because that interval length is close to that used in previous works whose focus was on the AIR-ENSO relationship. The 10-year sliding interval was chosen because it is half of the 20-year interval. Although the main conclusions of our results do not change if other interval lengths are chosen, we noted in the revised manuscript on Page 12 Lines 12 to 16 that other interval lengths were considered.

5. The authors have used correlation and sliding correlation to measure the relationship between the nonlinearity in ENSO and AIR anomalies, they could have very well used other measures such as mutual information. That would be more appropriate.

While the authors agree that there are other ways of measuring the association between two variables, correlation is the most frequent way AIR has been related to ENSO in the debate about the randomness of the AIR-ENSO relationship. As such, the authors feel that using the more common correlation method would allow researchers to more directly compare our results to those of previous works. Nevertheless, we added a discussion about mutual information and transfer entropy on Pages 16 and 17 of the revised

manusctipt to empathize that such method could prove useful for understanding the relationship between AIR and ENSO.

**6. The author have chosen AIR for the analysis, many studies have shown that the relationship between ENSO and indian rainfall is spatially variable, in that context how the application of AIR is justified and the results are meaningful.**

Although the authors agree that considering the spatial variability is necessary for all full understanding of the Indian monsoon, the inclusion of a spatial variability analysis is beyond the scope of the paper. This study is focused on the well-studied relationship between AIR and ENSO and contributes to the ongoing debate about the randomness of the AIR-ENSO relationship. However, because the omittance of a spatial variability analysis does represent an important limitation of the study, a dedicated discussion of the limitation was added to the discussion section of the revised manuscript on Page 16 Lines 25 to 34.

**Other minor comments, The author refer to the paper Schulte 2020 many times in the introduction, but I could not find it in the references Please correct the Eq.4 denominator.**

Thank you for the identifying the reference issue. The reference issue was corrected in the revised manuscript by correcting a reference entry in the reference list. The denominator of Eq. (4) was also corrected in the revised manuscript.

**References**

Carlos A. L. Pires & Abdel Hannachi (2021) Bispectral analysis of nonlinear interaction, predictability and stochastic modelling with application to ENSO, Tellus A: Dynamic Meteorology and Oceanography, 73:1, 1-30, DOI: 10.1080/16000870.2020.1866393

---

## Author Response (AR3)

The authors appreciate the below suggestions, which have been implemented in the revised manuscript. The suggestions are in bold text and our responses are in plain text.

**Comments to the Author:**

**The study and in particular the proposed method could be improved by a discussion about the sensitivity of the new skewness index on noise.**

The authors agree that understanding how noise influences the interpretation of waveform skewness is important. As such, we included a brief discussion about the impact of noise on Page 8 Lines 8 through 17 of the revised manuscript. The reader is now refereed to a supplementary figure that shows the result of an experiment that evaluates the impact of noise on waveform skewness.

**Moreover, in the introduction, the authors could also consider recent findings on additional triggers that force an increased link between ENSO and Indian monsoon rainfall:**

**Singh et al.: Fingerprint of volcanic forcing on the ENSO–Indian monsoon coupling, Science Advances, 6 (2020).**

**Maraun & Kurths: Epochs of phase coherence between El Niño/Southern Oscillation and Indian monsoon, Geophys Res Lett, 32 (2005).**

The authors appreciate the referral to these references, and we agree that a discussion about volcanic triggers is an important topic to discuss in the introduction section. We included a brief discussion about the role of volcanic forcing on the ENSO-Indian monsoon relationship on Page 2 Lines 25 through 31. The corresponding references have been added to the reference list.